# Use of Positron Emission Tomography for Pregnancy-Associated Cancer Assessment: A Review

**DOI:** 10.3390/jcm11133820

**Published:** 2022-07-01

**Authors:** Giulia Parpinel, Maria Elena Laudani, Francesca Paola Giunta, Chiara Germano, Paolo Zola, Bianca Masturzo

**Affiliations:** 1Unit of Gynecology and Obstetrics 2U, Sant’Anna Hospital, Città della Salute e della Scienza di Torino, University of Turin, 10126 Turin, Italy; giulia.parpinel@edu.unito.it (G.P.); melaudani@gmail.com (M.E.L.); paolo.zola@unito.it (P.Z.); 2Unit of Nuclear Medicine, Department of Medical Sciences, Città della Salute e della Scienza di Torino, University of Turin, 10126 Turin, Italy; fgiunta@cittadellasalute.to.it; 3Department of Obstetrics and Gynecology, Ospedale degli Infermi, 13875 Ponderano, Italy; bianca.masturzo@aslbi.piemonte.it

**Keywords:** positron emission tomography, cancer, pregnancy, dosimetry, ^18^F-FDG

## Abstract

**Background**. Positron emission tomography (PET) has proven clinical utility both in the initial and relapse staging phase, but this technique is controversial during pregnancy. The objective of this review is to provide a compendium of available information on the use of PET during pregnancy. **Materials and methods**. A systematic literature review was conducted from 1 January 2004 until 20 May 2021. A total of 4 small series and 9 case reports consisting of 25 cases were selected. **Results**. During the first trimester, the fetus is most sensitive to ionization damage, so lower doses are recommended (2.6E-02 mGy/MBq). Fetal-effective doses are higher in this period and the average fetal dose (4.06 ± 3.22 mGy) remains significantly below the threshold for deterministic effects. During the second and third trimesters, recommended doses are higher (1.4E-02 mGy/MBq at 6 months, and 6.9E-03 mGy/MBq at 9 months of gestation). ^18^F-FDG activity was distributed to the whole fetus with a prevalence of myocardial tissue in seven cases. The use of special precautions, such as PET-magnetic resonance (MR) and urinary bladder catheterization, reduces the amount of radioactive tracer. Breastfeeding interruption is not recommended. **Conclusions**. ^18^F-FDG PET is not contraindicated in pregnancy, but multidisciplinary discussion is necessary and strict precautions are recommended.

## 1. Introduction

Pregnancy-associated cancer is usually defined as cancer diagnosed during pregnancy or within the first year after delivery. The incidence of this condition, which is increasing and probably underestimated, has been estimated to be about 70–140 oncological diseases in every 100,000 pregnancies and 17–25:100,000 for cancer diagnosed during pregnancy [1]. The most common histotypes are malignant melanoma, breast, and cervical cancers. This presence of a malignant disease complicating the pregnancy requires the constant interaction of obstetric and oncological experts so that the choice of surgical, radiotherapy and chemotherapy treatments is compatible with pregnancy at different gestational ages. For this reason, this clinical condition is often considered difficult to manage and can have an impact on the management of pregnancy by increasing the tendency to anticipate the time of delivery and perform a caesarean section [2].

The presence of cancer during pregnancy requests an accurate initial staging of the primary lesion and metastases, which should consider the different nature of the primary neoplastic lesion and the compatibility of the examination with pregnancy. Commonly used instrumental examinations are ultrasound, computed axial tomography, magnetic resonance, and positron emission tomography (PET) imaging. The PET examination combines the anatomical information of computed tomography with the functional (metabolic) information from the administration of a specific radioactive tracer, most commonly ^18^F-Flourodeoxyglucose (FDG). ^18^F-FDG crosses the placental barrier and accumulates in fetal tissues [3,4,5,6], so knowing the fetal-absorbed dose is important for correctly assessing the risk to the fetus and for balancing this risk with the clinical benefit for the mother.

PET has proven clinical utility both in the initial and relapse staging phase, but the use of this technique is controversial during pregnancy and specific multidisciplinary counseling must be performed. Fetal radiation exposure resulting from medical imaging may be a complicated issue for pregnant women to understand. A lack of accurate and scientific data about the risks and benefits of ^18^F-FDG PET in oncological pregnant patients combined with medical liability concerns may result in inappropriate underestimation and consequently sub-treatment of the tumor or unjustified early termination of pregnancy. However, some authors have argued that PET/CT should not be absolutely contraindicated in pregnancy [7] and a recent qualitative survey of nuclear medicine physicians in Australia and New Zealand demonstrated that there is an appreciation for the use of PET/CT in some pregnant women with cancer in carefully selected clinical contexts [8]. 

The purpose of this review is to provide a compendium of available information on the use of PET in pregnant patients affected by malignant disease to support medical counseling and the management of these clinical conditions. 

## 2. Materials and Methods

A systematic literature review of PubMed and Google Scholar databases was conducted from 1 January 2004 until 20 May 2021 to identify the role of PET examination during pregnancy. We searched Medical Subject Headings (MeSH) terms in multiple combinations including positron emission tomography, cancer/oncological disease, pregnancy, radiation, fetal damage, obstetrical outcomes, absorbed dose, and 18-F-fluorodeoxyglucose. Studies included observational case reports and small series, textbooks and guideline statements. The research was restricted to human cases not involved in clinical studies without language restrictions applied. We selected only the cases or series where dosimetric data were available. The references from these papers were also analyzed. A total of 4 small series and 9 case reports were finally selected consisting of 24 patients and 25 cases because of the presence of a twin pregnancy.

## 3. Results

The summary of observational clinical studies examined in this review is presented in Table 1.

### 3.1. Fetal Radiation Dose and Potential Damages

*Short message.* Although it is not possible to define with certainty a threshold for deterministic effects in humans, it has been noted that radiation can interfere with cell proliferation after around 100 mGy in animals. The fetal radiation dose from ^18^F-FDG PET is significantly below this value: this suggests that the risk of radiation-related damage is relatively low if compared to maternal benefit.

The total fetal radiation dose is the sum of the doses derived from ^18^F-FDG and from transmission scans acquired for attenuation correction and additional exposure from radioactivity accumulating within the maternal organs. The first in vivo data were furnished by Benveniste et al. [5] who studied this process in nonhuman primates at the late stage of pregnancy in 2003. From this study, the current standard dosimetric values for ^18^F-FDG were derived: 2.2E-02 mGy/MBq for the first trimester of pregnancy and 1.7E-02 mGy/MBq after the first trimester. Pooling together subsequent different studies, the resulting range of doses in early pregnancy varies from 1.5E-02 to 4E-02 mGy/MBq. This difference can be explained by the fact that the first trimester is characterized by rapid cellular proliferation, which leads to a higher glucose consumption and consequently a high absorbed dose of ^18^F-FDG.

Some authors have described a potential risk of fetal radiation associated with the use of fluorine-18-labelled fluorodeoxyglucose PET/CT and thus discouraged the use of this exam during pregnancy [9,10]. The fetal radiation dose from ^18^F-FDG PET depends on the fetal weight, the type of radiotracer and the administered dose, but remains significantly below the threshold for deterministic effects due to exposure to ionizing radiation [7,11]. Based on the data derived from the study of radiation damage in animals [12], it has been noted that radiation can interfere with cell proliferation after relatively low doses, around 100 mGy. Although it is not possible to define with certainty a threshold for deterministic effects, as it would not be ethical to conduct this type of experiment on humans, it can be noted that the dose of radiation to which the fetus is exposed in utero is much less than 100 mGy. During 8–15 weeks of gestation, radiation exposure might induce non-stochastic effects such as embryonic death, malformations, growth retardation and microcephaly at a threshold between 0.35 and 0.5 Gy [13]. For a fetus of 16–40 weeks, radiation exposure is associated with non-stochastic effects of growth defects, mental retardation or reduced brain dimensions at a threshold of 1.5 Gy. The fetal nervous system exhibits a long period of sensitivity to ionizing radiation during the whole gestation period, and its development is known to be affected by radiation exposure above 0.5 Gy [14].

Stochastic effects have also been postulated: for example, radiation-induced carcinogenesis [15]. Based on the hypothesis of mutagenic agents, this kind of effect does not have a specific threshold. This does not mean that the risk does not increase as the exposure increases, but that the unfavorable outcomes have the same severity in an exposed population as in a non-exposed population. Mental disability has also been proposed as a possible stochastic effect, although there is no agreement among authors. In addition, a 2008 study [16] found that the risk of developing stochastic effects after in utero exposure may be lower than the risk of early childhood exposure. Therefore, the literature data about this topic are insufficient and often conflicting, thus not allowing the quantification of risk, especially if compared with the other risks of pregnancy [17]: a 3–4% risk of congenital malformations and a 15% chance of miscarriage attended in physiological pregnancies.

Fetal dose estimates for ^18^F-FDG were originally calculated with no information on placental crossover and fetal uptake [18] and later modified using placental crossover data observed in primates [19]. First, biological distribution of a tracer varies significantly in the presence of pathological conditions. Second, in the presence of a pregnant patient, the calculation is complicated by the stage of pregnancy and the change in the body’s functions/metabolic processes. These doses are obtained with disparate methodologic approaches: Monte Carlo code and simple geometric models, using the dose to the uterus as a proxy, referring to different assumptions on the kinetics of the mother’s bladder. The average dose to the fetus across the literature reviewed is 4.06 ± 3.22 mGy [20].

Xie et al. [21] developed a complete set of embryo/fetus models at various gestation periods with 25 identified tissues according to reference data recommended by the ICRP publication 89 [22], representing the anatomy of the developing embryo/fetus. Firstly, they demonstrated that the fetal-effective doses are significantly higher in the first trimester than at other gestational periods because the fetal weight is lower at the beginning of pregnancy. Secondly, this dose is nonuniformly distributed in the fetus and changes with the different week of gestation: at 8 weeks, fetal kidney and liver receive the highest dose from ^18^F-FDG, while above 10 weeks of gestation, the bone marrow, brain, and thyroid receive the highest absorbed radiation doses. Thirdly, the component of fetal-absorbed dose from the maternal body decreases with gestational age while the self-absorbed dose of the fetus increases. Based on the phantom series proposed by Stabin and the Society of Nuclear Medicine and Molecular Imaging RAdiation Dose Assessment Resource (RADAR) task force [23], Zanotti-Fregonara et al. [24] recently published recommended fetal doses: 2.6E-02 mGy/MBq in early pregnancy, 1.9E-02 mGy/MBq at 3 months of gestation, 1.4E-02 mGy/MBq at 6 months of gestation, and 6.9E-03 mGy/MBq at 9 months of gestation. Activity coefficients reach a plateau after 34 weeks.

For all the 25 cases examined in this review, a description of nuclear data (imaging method, administered activity, CT absorbed dose, PET absorbed dose, total absorbed dose, fraction and time integrated activity), neonatal and obstetrical data (gestational age, weight, fetal FDG-avid structures and outcomes) are reported in Table 2.

### 3.2. PET during the First Trimester

*Short message.* There are limited data available about the possible fetal damages after the execution of PET during the first trimester of pregnancy. However, after examining the literature where these data are explicitly reported, we did not find specific mentions of unfavorable fetal outcomes.

The embryogenic phase ranges from conception to 12 weeks of gestation. During this critical period, environmental insults can have serious consequences for the development of the embryo and outcomes can be variable according to the different weeks of gestation. As postulated by Bergonie and Tribondeau [25], the radiosensitivity of living cells is directly proportional to their rate of division and inversely related to their degree of specialization. Weeks 3 and 4 are the most sensitive for abortion, which might occur after irradiation with 100 mGy, while the period between weeks 4 and 15 is the most sensitive for growth decline, microcephaly and mental disability, which occur at doses higher than 200 mGy according to Brent et al. [15]. However, there are probably no deterministic effects with radiation exposure soon after conception, and the threshold dose for deterministic effects is higher during the early pregnancy or embryonic period. Fetal-absorbed doses in early pregnancy have been suggested to be higher than standard values in some studies, but this opinion is not shared by all authors. The available data on the use of PET during pregnancy in this period are scarce and the general attitude is to postpone PET until after 12 weeks of gestation.

The first examples of the use of PET during the first trimester of pregnancy are isolated case reports of patients who assessed the oncological response to therapy without knowledge of their pregnancy status [26]. From our review of the literature, we listed only six cases of patients who performed PET imaging during the first trimester [3,4,27,28]. In most of cases, the exam was a PET/CT (five cases out of six); one case of PET only was reported by Zanotti-Fregonara et al. [28] to assess the stadium of a metastatic breast cancer. The mean injected dose was 371.7 MBq (range 296–583 MBq). As reported by Takalkar et al. [27], when pregnancy is confirmed in every trimester, the dose of radiotracer is normally reduced to 173.9–340.4 MBq. In this series, the patient who received the highest dose of ^18^F-FDG (583.12 MBq) performed PET-CT by accident, so the standard protocol was applied.

The total absorbed dose is the sum of the CT and PET absorbed dose. In our research, the CT absorbed dose was reported in only two cases (8.3 and 10 mGy), while considered negligible in two cases, and finally not reported in two cases. An average dose of 7 mGy (range 2.7–11.8 mGy) was attributed to PET and a mean total dose of 13.11 mGy (range 2.7–21.8 mGy) was calculated, excluding two cases when the doses were not mentioned. It is confirmed in the literature that the ^18^F-FDG fetal uptake in early pregnancy is higher than current dosimetric standards ranging from 7.25E-03 to 1.73E-02 mGy/MBq after 1 h voiding [28]. Another recent study outlines that an administered maternal dose of 200 MBq would deliver a fetal dose ranging from 5.2 mSv in early pregnancy to 1.4 mSv in late pregnancy. This phenomenon could be explained by the smaller volume of the fetus and by undifferentiated and rapidly proliferating cells that compose the fetal body at this stage of pregnancy.

In our study, only two authors reported fetal outcomes, specifying that pregnancies ended with healthy newborns. An interesting observation regarding the distribution of ^18^F-FDG outlined that while early in the trimester the whole fetus maintains the same avidity for the tracer, later in the trimester some structures such as the myocardium, kidneys and bladder are the most affected by this phenomenon, as is described to happen in the following trimesters.

### 3.3. PET during the Second and Third Trimesters

*Short message.* During the second and third trimesters of pregnancy, the total dose of radiation absorbed by the fetus reported in the literature is lower than in the first trimester, and in this case no fetal damages have been registered. The structures with the highest tracer accumulation were found to be the urinary myocardium, brain, and urinary system.

During the second and third trimesters, PET is generally performed after the patient has signed informed consent, but there are rare cases in the literature where pregnancy has been diagnosed through images of this diagnostics [29]. This examination is fundamental for the correct diagnosis and staging of the disease that affects the modalities and timing of childbirth. The risk of a mis-staged carcinoma appears greater than the fetal risk of an optimized low-dose PET/CT procedure at this conception age [30].

Our review of the literature is based on 19 patients who underwent PET imaging during this period of pregnancy [20,27,28,30,31,32,33,34,35,36]. In one case [36], data of the same pregnancy are reported twice because of a twin gestation. In most of the cases (16 out of 19), PET/CT was performed; two cases performed PET/MRI and one case PET only. The distribution of oncological diagnosis reflected the literature distribution: 42% hematological diseases (eight cases), 15% breast cancer (three cases), 10% cervical cancer (two cases), 21% other tumors (one glioma, one pheochromocytoma, one melanoma, one parathyroid cancer). In two cases, these data were not available. The mean gestational age was 25.47 weeks (range 18–34) and the average maternal age, when available, was 32.2 years.

The average injected dose turned out to be 259.12 MBq (range 146–555), significantly lower than the dose administered in the first trimester of pregnancy. Consequently, lower total absorbed doses were reported: the mean total absorbed dose calculated in 16 out of 19 cases was 3.97 mGy (range 1.01–14.4), with a CT-absorbed dose of 1.80 mGy (range 0–6) and PET-absorbed dose of 2.89 mGy (range 6.4–1.01). Similar not elevated values of fraction- (average 0.136; range 0.00086–0.052) and time-integrated activity (average 0.036 Bq-h/Bq; range 0.0023–0.137) were reported. Fetal self-dose was reported to progressively increase with the size of the fetus [36], ranging from about 10% during the first trimester to about 40% at the end of pregnancy. The value found in early pregnancy (2.5E-02 mGy/MBq) is comparable to the standard value estimated from monkeys [19], while for the remaining duration of the pregnancy, the values from human data are about half to one-third of those estimated from monkeys. The placental contribution to the final dose to the fetus is small, ranging from about 1.8% in the second trimester [30] to 1.2% in the first trimester, probably due to the larger volume of the placenta in the second trimester.

Reviewing the literature, ^18^F-FDG activity was distributed to the whole fetus with a prevalence of myocardial tissue in seven cases, corresponding to more early gestations (<26–28 weeks), while after that period the myocardial and brain areas were the most active ones. Blanc-Durand et al. [35] interestingly observed, in their 30-weeks-pregnant case report, a low fetal brain uptake and symmetrical myocardial metabolism, probably due to the specific fetal circulation with high lung pressure and ductus arteriosus bypass, which make pressure in the left ventricles identical. Similarly, in the largest series published [36], the rate of ^18^F-FDG uptake of the brain was low, comparable to that in the soft tissues. The authors attributed that finding to the low metabolic status of this organ in the uterus, as well as in newborns.

### 3.4. PET and Pregnancy: How to Reduce Fetal–Neonatal Injuries?

*Short message.* The administration of ionizing radiation can be reduced using PET/MRI instead of PET/CT and an adequate oral or intravenous hydration before and during the examination. Breastfeeding interruption is not recommended, but external breast milk collection and bottle-feeding by a third person should minimize radiation exposure to the infant.

While with PET-only scans, the absorbed dose from radioactive external sources is negligible [37], the calculation of the total dose must consider the fact that CT-PET scans are performed with the advantage of better image definition and the disadvantage of adding 6–14 mGy. PET-MRI avoids the administration of ionizing radiation, and the value of added irradiation is likely 5 mGy or less, so it could be the best choice for pregnant patients. A recent study has shown that whole-body diffusion-weighted MRI could replace ^18^F-FDG PET/CT as it presents equal efficacy in the detection of nodal and distant metastasis, including bone metastasis, both in solid tumors and lymphomas [38].

The tracer is excreted by the kidneys and maintaining adequate oral or intravenous hydration before and during the assessment is essential to reduce radiation exposure. Due to the anatomic position with respect to the fetus, the bladder is the main determinant of radiation, and rapid elimination of the radioactive urine from the urinary bladder can facilitate minimizing fetal radiation exposure. One-hour voiding time is the most likely voiding schedule, especially when the pregnancy is unknown [36]. Urinary bladder catheterization can continuously drain the urine, thus reducing the amount of radioactive tracer. Some authors [27] also add to this protocol the use of 10 mg Furosemide, administered intravenously 15 min after ^18^F-FDG administration to force diuresis [39].

Many radiopharmaceuticals are also known to be excreted in breast milk and their activity can be directly measured. Their presence in breast milk and the consequent accumulation in the organs of the newborn could cause long-term effects on the newborn’s development of neonatal organs and anatomy. High concentrations of ^18^F-FDG in lactating breast have been reported [40]. Despite being based on the study of only six patients, Hicks et al. [41] reported a decay-corrected activity measurable in breast milk of 5.54–19.3 Bq/mL/MBq injected and subsequently a calculated maximum cumulative dose to the infant of 0.085 mSv. This dosage was below the recommended limit of 1 mSv. The main source of potential radiation is more related to the proximity of the infant to the maternal breast rather than to the ingestion of milk. For this reason, no interruption of breastfeeding is recommended in patients undergoing ^18^F-FDG assumption [17], but external breast milk collection and bottle-feeding by a third party should be performed to minimize radiation exposure to the infant.

## 4. Conclusions

This is, to our knowledge, the first monographic review of the literature about the use of PET during pregnancy with the largest number of cases reported. From this systematic review of the literature, we can postulate that ^18^F-FDG-PET during pregnancy could be proposed to oncological patients in every trimester of pregnancy. Since the fetal total absorbed dose is significantly lower than the suggested threshold of deterministic effects in animals, it is not yet possible to attribute to this examination a negative effect principally regarding fetal malformations, while the advantage for the mother in terms of the staging and diagnosis of the recurrence of neoplastic disease is clearly in favor of using this examination. However, it should be emphasized that the literature data are generally scarce and that there are not enough data on stochastic and long-term effects. Future studies should provide more information about the follow-up of these children, possibly through the creation of an international registry. To conclude, ^18^F-FDG-PET is not contraindicated in pregnancy, but a multidisciplinary approach is strongly indicated to evaluate the clinical situation and other diagnostic options; moreover, specific precautions must be taken, such as hydration and repeated urinary bladder voiding [40].

## Figures and Tables

**Table 1 jcm-11-03820-t001:** Summary of observational clinical studies on PET examination use during pregnancy.

First Author	Year of Publication	Number of Patients	Type of Cancer
Townson	2004	2	Glioma
			Hodgkin’s disease
Zanotti-Fregonara	2008	1	Hodgkin’s disease
Zanotti-Fregonara	2010	1	Hodgkin’s disease
Takalkar	2011	6	Lung cancer
			B-cell non-Hodgkin lymphoma
			B-cell non-Hodgkin lymphoma
			Cervical cancer
			B-cell non-Hodgkin lymphoma
			Cervical cancer
Zanotti-Fregonara	2012	1	Pheochromocytoma
Hsieh	2012	1	Breast cancer
Calais	2014	1	Hodgkin’s disease
Erdogan	2015	1	Hodgkin’s disease
Zanotti-Fregonara	2015	6	Vocal cord
			Cervical cancer
			Breast cancer (metastatic)
			Hodgkin’s disease
			Burkitt lymphoma
			Melanoma
Zanotti-Fregonara	2016	1 (twins)	NA
			NA
Gill	2018	1	Parathyroid cancer
Blanc-Durand	2019	1	Breast cancer
Drouet	2020	1	Breast cancer

Abbreviations: NA, not available.

**Table 2 jcm-11-03820-t002:** Characteristics of patients, nuclear dosimetric data, fetal FDG-avid structures and outcomes.

Patient	Imaging Method	Gestational Age (Weeks)	Weight (kg)	Administered Activity-Injected Dose (MBq)	CT Absorbed Dose (mGy)	PET Absorbed Dose (mGy)	Total Absorbed Dose (mGy)	Fraction	Time Integrated Activity (Bq-h/Bq)	Fetal FDG-Avid Structures	Fetal Negative Outcomes
1	PET/CT	26	/	188	~0	3.2	3.2	/	/	Head and body	ND
2	PET/CT	22	/	355	6	6.4	12.4	/	/	Whole fetus	ND
3	PET/CT	8	60	320	8.3	10.6	18.9	0.0020	0.0053	Whole fetus	No
4	PET/CT	10	71	296	10	11.8	21.8	0.0018	0.0046	Whole fetus	NA (Therapeutic interruption)
5	PET/CT	6	67.6	583	~0	9.04	9.04	0	0	Whole fetus	No
6	PET/CT	18	87.8	200	~0	1.43	1.43	0.00086	0.0023	Whole fetus Myocardium	No
7	PET/CT	25	67.1	337	~0	2.1	2.1	0.0084	0.0223	Whole fetus Myocardium	No
8	PET/CT	28	81.8	174	~0	1.01	1.01	0.0071	0.0187	Whole fetus Myocardium	No
9	PET/CT	30	88.6	229	~0	2.43	2.43	0.0196	0.0518	Myocardium	No
10	PET/CT	23	58.9	181	~0	1.1	1.1	0.0078	0.0206	Whole fetus Myocardium	No
11	PET/CT	21	53	181	3.6	5	8.6	0.0049	0.0129	Whole fetus, Brain	No
12	PET/CT	26	/	370	3.6	6.29	9.89	/	/	Myocardium Kidneys	ND
13	PET/CT	26	/	ND	3.2	4.1	7.3	/	/	Myocardium	No
14	PET/CT	34	95	555	/	/	/	0.0192	0.0507	ND	ND
15	PET/CT	5	85.8	296	ND	5.1	NA	0.00115	0.003	Myocardium Kidneys Bladder	ND
16	PET/CT	12	58.1	385	ND	2.8	NA	0.00061	0.0016		ND
17	PET	~12	77.1	350	~0	2.7	2.7	0.001	0.0026		ND
18	PET/MR	19	69.9	296	NA	1.4	1.4	0.00366	0.0097		ND
19	PET/MR	19	50.8	348	NA	1.4	1.4	0.00238	0.0063		ND
20	PET	28	65.8	296	~0	1.8	1.8	0.01951	0.0515		ND
21 (twins)	PET/CT	25	76	188	/	/	/	0.0156	0.0412	ND	ND
	PET/CT	25	76	188	/	/	/	0.0164	0.0434	ND	ND
22	PET/TC	28	/	173	3.1	2.8	5.9	/	/	Myocardiu Kidneys, Bladder Head	No
23	PET/TC	30	/	ND	/	/	1.9	/	/	Myocardium	No
24	PET/CT	31	/	146	3.9	/	1.7	0.052	0.137	ND	No

Abbreviations: PET, positron emission tomography; CT, computed tomography; MR, magnetic resonance; FDG, 18-F-fluorodesossyglucose; ND, not described.

## Data Availability

Not applicable.

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
