# Peer review of "Use of Positron Emission Tomography for Pregnancy-Associated Cancer Assessment: A Review"

_jcm, 2022, doi:10.3390/jcm11133820_

Round 1

Reviewer 1 Report

Important and interesting manuscript on the radiation safety concerns with the use of PET on cancer diagnostics during pregnancy.

The manuscript is well written, nevertheless, a discussion about the uncertainty considering possible stochastic effects on the fetus is mandatory and should be added. Seeing that even with the review we look at data of 25 unborns only the conclusion should be written with more caution. Furthermore, the conclusion should mention the limited data and the need for follow-up on these patients. Maybe even an international register.

In detail:

line 83/84: the threshold for deterministic effects: change the wording. A threshold cannot be set in my opinion - it might be assumed or there might be hints - the threshold discussion is very complicated. In this manuscript, it looks like clear evidence - which I don`t see in such a simplified way.

Furthermore, in reference 12 I didn`t find a passage setting the threshold to 100 mSv - for answering the review please provide a clear hint to where to find this data.

line 86: damage acceptable? There are no data which prove this assumption and there is no discussion on late effects, therefore, this statement is not warranted - change wording.

line 110: cGy: staying within the regular units Gy and mGy is much easier for the reader

line 112: negligible - are the authors able to quantify the stochastic risks after these fetal absorbed doses for childhood/adult cancers or even noncancer effects? If not wording like "negligible" should be omitted.

line 147: again - there are just not enough data - please more caution with the wording

Table 2: wrong use of decimal separator

Conclusions: Description of very limited data and missing data on stochastic effects is missing.

References 12 line 317. The first author of this publication is Streffer and some more authors. Jack Valentin is the editor - please check with the Reference style.

Author Response

Reviewer 1

Important and interesting manuscript on the radiation safety concerns with the use of PET on cancer diagnostics during pregnancy.

The manuscript is well written, nevertheless, a discussion about the uncertainty considering possible stochastic effects on the fetus is mandatory and should be added. Seeing that even with the review we look at data of 25 unborns only the conclusion should be written with more caution. Furthermore, the conclusion should mention the limited data and the need for follow-up on these patients. Maybe even an international register.

Comment: Concerning data about stochastic effects, the text has been modified as follows: Also stochastic effects have been postulated: for example radiation induced carcinogenesis [15]. Based on the hypothesis of mutagenic agents, this kind of effect doesn’t have a specific threshold. It doesn’t mean that the risk doesn’t increase as the exposure increases, but that the unfavorable outcomes have the same severity in exposed population as in non-exposed population. Mental retardation has also been proposed as a possible stochastic effects, although there is no agreement between the authors. In addition, a 2008 study [16] found that the risk of developing stochastic effects after in utero exposure may be lower than the risk of early childhood exposure. Therefore, the literature data about this topic are insufficient and often conflicting to allow the quantification of risk especially if compared with the other risks of pregnancy [17]: 3-4% risk of congenital malformations and a 15% chance of miscarriage attended in physiological pregnancies.

Moreover, two references (reference 15 and 16) have been added to the text.

In detail:

  1. line 83/84: the threshold for deterministic effects: change the wording. A threshold cannot be set in my opinion - it might be assumed or there might be hints - the threshold discussion is very complicated. In this manuscript, it looks like clear evidence - which I don`t see in such a simplified way.

Comment: Text has been modified to be more adherent to the literature informations: Although it is not possible to define with certainty a threshold for deterministic effects in humans, it has been noted that radiation can interfere with cell proliferation after around 100 mGy in animals. The fetal radiation dose from 18F-FDG PET in significantly below this value: this suggests that the risk of radiation-related damage is relatively low if compared to maternal benefit.

  1. Furthermore, in reference 12 I didn`t find a passage setting the threshold to 100 mSv - for answering the review please provide a clear hint to where to find this data.

Comment: In reference 12 is reported:

  • Chapter 2, page 11-12: Radiation effects during the developmental period have mainly been observed in mice and rats, but data from rabbits and dogs are also available. Radiation can interfere with cell proliferation after comparatively small doses (around 100 mSv and below during sensitive periods, especially the zygote stage) and this leads to a delay of development; cells will die and the death of the whole embryo may follow.
  • Chapter 6, page 132: This supports the conclusion of UNSCEAR(2000), based on data on tumour induction byirra diation after birth, that for most tumour types in experimental animals, a significant increase in risk is detectable only at doses of above about 100 mGy.

Text has been modified to be more adherent to the literature informations: Based on the data derived from the study of radiation damage in animals [12], it has been noted that radiation can interfere with cell proliferation after relatively low doses, around 100 mGy. Although it is not possible to define with certainty a threshold for deterministic effects, as it would not be ethical to conduct this type of experiments on humans, it can be noted that the dose of radiation to which the fetus is exposed in utero is much less than 100 mGy.

  1. line 86: damage acceptable? There are no data which prove this assumption and there is no discussion on late effects, therefore, this statement is not warranted - change wording.

Comment: acceptable has been eliminated

  1. line 110: cGy: staying within the regular units Gy and mGy is much easier for the reader

Comment: 50 cGy has been corrected to 0.5 Gy

  1. line 112: negligible - are the authors able to quantify the stochastic risks after these fetal absorbed doses for childhood/adult cancers or even noncancer effects? If not wording like "negligible" should be omitted.

Comment: text has been modified as follows: Therefore, the literature data about this topic are insufficient and often conflicting to allow the quantification of risk especially if compared with the other risks of pregnancy

  1. line 147: again - there are just not enough data - please more caution with the wording

Comment: text has been modified as follows: There are few data available about the possible fetal damages after execution of PET during the first trimester of pregnancy. However, after examining literature where these data are explicitly reported, we didn’t find specific mentions to unfavorable fetal outcomes.

  1. Table 2: wrong use of decimal separator

Comment: comment accepted, data correction

  1. Conclusions: Description of very limited data and missing data on stochastic effects is missing.

Comment: text has been integrated as follows: This is to our knowledge the first monographic review of literature about the use of PET during pregnancy with the largest number of cases reported. From this systematic review of literature, we can postulate that 18F-FDG-PET during pregnancy could be proposed to oncological patients in every trimester of pregnancy. Since fetal total absorbed dose is significantly lower than the suggested threshold of deterministic effects in animals, it is not yet attributable to this examination a negative effect principally regarding fetal malformations, while the advantage for the mother in terms of staging and diagnosis of recurrence of neoplastic disease is clearly in favor of the use of this exam. However, it should be emphasized that literature data are generally scarce and that there are not enough data on stochastic and long-term effects. Future studies should provide more informations about the follow-up of these children, possibly through the creation of an international registry. To conclude, 18F-FDG-PET is not contraindicated in pregnancy, but a multidisciplinary approach is strongly indicated to evaluate the clinical situation and other diagnostic options; moreover, specific precaution must be taken such as hydration and repeated urinary bladder voiding [41].

  1. References 12 line 317. The first author of this publication is Streffer and some more authors. Jack Valentin is the editor - please check with the Reference style.

Comment: comment accepted, Reference Style correction

Reviewer 2 Report

Well researched and organized manuscript discussing a little researched topic. Bringing attention to the relatively low risks of PET studies in pregnant cancer patients may certainly help ensure they are given proper workups and assist with clinical decision making. This manuscript compiles relatively rare case reports into a central, well organized format.

The section on reducing fetal injuries is important guidance for clinicians.

Author Response

Reviewer 2

Well researched and organized manuscript discussing a little researched topic. Bringing attention to the relatively low risks of PET studies in pregnant cancer patients may certainly help ensure they are given proper workups and assist with clinical decision making. This manuscript compiles relatively rare case reports into a central, well organized format.

The section on reducing fetal injuries is important guidance for clinicians.

Comment: no correction requested

Round 2

Reviewer 1 Report

All points raised have been adressed by the authors. Thank you for answering point to point.